# Ultrasound-Assisted Enzymatic Protein Hydrolysis in Food Processing: Mechanism and Parameters

**DOI:** 10.3390/foods12214027

**Published:** 2023-11-04

**Authors:** Jingya Qian, Di Chen, Yizhong Zhang, Xianli Gao, Ling Xu, Guoqiang Guan, Feng Wang

**Affiliations:** 1School of Food and Biological Engineering, Jiangsu University, Zhenjiang 212013, China; sjqjy@126.com (J.Q.); 2212218011@stmail.ujs.edu.cn (D.C.); 2221718038@stmail.ujs.edu.cn (Y.Z.); gaoxianli@ujs.edu.cn (X.G.); lxu@ujs.edu.cn (L.X.); ggqyxq@ujs.edu.cn (G.G.); 2Institute of Agricultural Products Processing Engineering, Jiangsu University, Zhenjiang 212013, China

**Keywords:** ultrasound pretreatment, enzymatic hydrolysis, mechanisms, application, ultrasound devices

## Abstract

Ultrasound has been widely used as a green and efficient non-thermal processing technique to assist with enzymatic hydrolysis. Compared with traditional enzymatic hydrolysis, ultrasonic-pretreatment-assisted enzymatic hydrolysis can significantly improve the efficiency of enzymatic hydrolysis and enhance the biological activity of substrates. At present, this technology is mainly used for the extraction of bioactive substances and the degradation of biological macromolecules. This review is focused on the mechanism of enzymatic hydrolysis assisted by ultrasonic pretreatment, including the effects of ultrasonic pretreatment on the enzyme structure, substrate structure, enzymatic hydrolysis kinetics, and thermodynamics and the effects of the ultrasonic conditions on the enzymatic hydrolysis results. The development status of ultrasonic devices and the application of ultrasonic-assisted enzymatic hydrolysis in the food industry are briefly described in this study. In the future, more attention should be paid to research on ultrasound-assisted enzymatic hydrolysis devices to promote the expansion of production and improve production efficiency.

## 1. Introduction

Ultrasound (US) is defined as sound waves above the upper limit of audibility for the human ear (from 16 Hz to 20 kHz) [1]. Ultrasonic systems are classified into different categories based on the frequency of the sound waves based on the mode of application and also based on the geometry of the emitter. Based on the sonic frequency, high- and low-frequency US exists [2]. Low-frequency US (from 100 kHz and above) is mainly used for high-precision detection within food processing, medical, and diagnostic usages. High-frequency US (from 20 to 100 kHz), as a green, efficient, safe, and novel physical processing technology, has been widely used in the food processing industry [3,4]. The ultrasonic cavitation effect is usually described as the process by which bubbles in a liquid medium form, grow, and collapse out of control under the action of an ultrasonic field of sufficient pressure [5,6]. The cavitation effect caused by ultrasound can be divided into steady-state cavitation and transient cavitation [7]. Steady-state cavitation occurs when the volume of the liquid hollow bubble gradually expands under continuous high-frequency ultrasonic conditions but does not exceed the critical crack size. During this process, the micro-flow generated by the change in the volume of the cavitation bubble creates strong eddy currents in the medium surrounding the cavitation bubble. At the same time, the diffusion of dissolved gas into and out of the bubble in the medium also generates a small current around the bubble. Transient cavitation means that, under the condition of continuous low-frequency ultrasound, the volume of the bubble expands rapidly, and after reaching the critical size, the bubble bursts and is accompanied by a large amount of energy release. Ultrasonic transient cavitation creates high shear stresses and temperatures around the bubble instantaneously and can lead to the production of hydrogen and hydroxyl radicals [8]. The expansion, contraction, explosion, and implosion of bubbles in fluid media may produce extreme conditions and play a leading role in ultrasonic action [9,10]. The sonic cavitation induced by ultrasound will produce a large number of microbubbles in aqueous solution, and the collapse of microbubbles will produce a variety of active intermediates (OH, HO_2_·), causing mechanical and thermal effects (e.g., blast waves, high temperatures, micro-jets, and acoustic streaming) [11,12]. Overall, acoustic cavitation plays a crucial role in the application of ultrasonic food. It is feasible to significantly improve the ultrasonic effect by adjusting the ultrasound generator parameters to the optimal cavitation conditions.

During food processing, ultrasound-based techniques can be used for diverse applications, such as extraction, sterilization, drying, freezing, fat separation, pasteurization, controlling lipid oxidation, homogenization, emulsification, disinfection, protein denaturation, enzyme activation or inactivation, etc. [13]. Compared with traditional extraction methods, ultrasound-assisted extraction has the advantages of a high product yield, short extraction time, low solvent consumption, and low environment pollution, so it could be used to extract high-value-added compounds from fruit residues and agricultural products, such as the extraction of pectin and bioactive substances [14,15,16]. Ultrasound has demonstrated its ability to enhance the preservation of important nutrients that are present in fruit and vegetable juices [17]. Studies have shown that, compared with pasteurized juice, ultrasonic-treated kutkura juice has a higher retention rate in terms of the total phenol content and ascorbic acid content [18]. Ultrasonic and enzyme extraction can significantly reduce the particle size of Noni juice and improve the stability and rheological properties of the suspension. Noni juice exhibits high contents of total phenolics and flavonoids, 148.19 ± 2.53 mg gallic acid/100 mL and 47.19 ± 1.22 mg rutin/100 mL, respectively, thus contributing to better antioxidant activity [19]. Because no chemicals are added and it can lower the formation of toxic compounds [20], ultrasound can effectively reduce the number of harmful microorganisms in food, such as reducing *Listeria* and *Pseudomonas* spp. in Chinese cabbage [21,22] and inactivating or eliminating contaminated bacteria in semi-skimmed sheep milk [23]. Moreover, ultrasound was also shown to remove some phytoviruses in fruits [24]. Due to the ability to enhance the mass transfer process in liquid and gas environments and then produce increases in the effective moisture diffusivity and mass transfer coefficient, ultrasound can assist with intensifying the drying process, such as shortening the drying time, reducing the drying temperature, and saving energy consumption [25,26,27,28]. Meanwhile, ultrasound can also enhance the freezing process due to its abilities to initiate nucleation, increasing the heat- and mass transfer rates, and controlling the crystal shape and size distribution in frozen foodstuffs [29,30]. An ultrasonic power density of 20.5 W/L can significantly improve the gluten network structure of frozen dough, making it more orderly and complete [31]. Appropriate ultrasound power treatment could improve the freezing quality and freezing speed of porcine longissimus muscles [32]. In addition to these applications, high-intensity ultrasound can be considered an effective emulsification technique. It could be used to produce stable emulsions of soybean protein isolate [33], even producing a stable human milk fat analog emulsion using protein and lecithin at low levels [34]. Ultrasonic is also a reliable method for the nondestructive detection of moisture and salt in ham [35]. Ultrasound can also improve the taste and flavor of foods. Chicken broth prepared via ultrasonic and NaHCO_3_ pretreatment has a darker color, lower pH value, and higher contents of umami nucleotides and amino acids, which can enhance the umami and saltiness of chicken broth and inhibit its bitterness and astringency [36]. 

Enzymatic hydrolysis is a process in which enzymes facilitate the cleavage of bonds in molecules with the addition of the elements of water. Enzymatic hydrolysis technology has the advantages of a simple reaction, mild reaction conditions, high reaction efficiency, and no, or very few, adverse reactions. Enzymatic hydrolysis technology is widely used in food processing, such as for extracting polyphenols and reducing sugars and assisting in the fermentation of fruit wine [37,38,39]. Research shows that enzymolysis could be used to produce low-molecular-weight bioactive peptides, such as highly active angiotensin converting enzyme inhibitory peptides [37,39]. However, traditional enzymolysis has many shortcomings, which include a low contact frequency with substrates, a long enzymolysis time, decreased enzyme activity, and the aggregation of substrates [40]. Therefore, many methods have been investigated in order to solve these problems during enzymolysis, such as the ultrasound-assisted enzymatic system. Ultrasound is an emerging technology that produces acoustic waves, which cause the rapid formation and collapse of bubbles. It has the capacity to break hydrogen bonds and interact with polypeptide chains due to Van der Waals forces, leading to the alteration of the secondary and tertiary structure of the enzymes, thereby leading to a loss in their biological activity [2].

Recently, there have been many reports on the application of ultrasound-assisted enzymatic hydrolysis [41,42,43,44,45,46]. In addition, some authors have reported the mechanism of enhanced enzyme activity due to ultrasound in the form of a review [47,48,49]. In particular, the positive role of ultrasound was reviewed in an enzymatic process from the aspects of ultrasound-modified enzymes, ultrasound-assisted immobilization, ultrasound pretreatment, and ultrasound-assisted enzymatic reactions [50]. In addition, based on the comprehensive effects of thermal and mechanical actions generated by the acoustic cavitation of ultrasonic waves, which generally helps to improve the mass transfer efficiency and the diffusion of substrates and the enzyme cover disorientation of the configuration of the substrate [46]. The objectives of this paper are to comprehensively summarize the mechanism, key parameters, and various ultrasonic devices used for ultrasound-assisted enzymatic hydrolysis and to introduce the application of ultrasound-assisted enzymatic hydrolysis in food processing. We used the Google Scholar, Web of Science, and Elsevier Scopus search engines to summarize reviews and articles produced between 2010 and 2023 using ultrasonic/ultrasound, enzymatic hydrolysis, mechanism, application, and ultrasound devices as keywords. The present review provides an overview of the changes in the surface properties and structures of substrates when ultrasonic pretreatment is introduced, and the effects of ultrasonic pretreatment conditions on enzymatic protein hydrolysis are briefly discussed. Moreover, the effects of ultrasound-enzyme synergistic treatment on the enzyme structure and substrate structure are summarized, the key parameters of the synergistic treatment are revealed, and the application of ultrasound-assisted enzymatic hydrolysis in food processing is summarized. Finally, the related ultrasonic devices are compared.

## 2. Ultrasonic Pretreatment before Enzymatic Hydrolysis

### 2.1. Mechanism of Ultrasonic Pretreatment

Ariana de Souza Soares first studied the effect on invertase. It was found that ultrasonic treatment did not change the structure of the substrate [51]. However, in numerous studies, ultrasound has had a great impact on enzyme substrates, so it is necessary to study the impact of ultrasound on substrates.

#### 2.1.1. Effect of Ultrasonic Pretreatment on the Surface Properties of Substrates

Ultrasound has attracted widespread attention in the food industry and technology field due to its valid effect. Ultrasonic pretreatment, as a pretreatment method, has significant influences on the promotion of enzymolysis processing, such as by increasing the binding of enzymes to substrates by changing the surface morphology of substrates (Figure 1). Huang et al. [52] used scanning electron microscopy (SEM) to characterize the morphology and structure of insoluble dietary fiber (IDF) from garlic stalks. They reported that the surface of the IDF pretreated with ultrasound had a honeycomb-like structure with more cracks and pores, while the surface of the sample untreated by ultrasound was flat. Similarly, according to the SEM analysis, Zhong et al. [53] found that straw treated with ultrasound-assisted tetrabutylammonium hydroxide was more relaxed, dispersed, and had greater surface fiber porosity than straw treated with only tetrabutylammonium hydroxide. In addition, Alondra M. Idrovo Encalada et al. [54] found that cell wall thickness can be clearly discovered in non-US treated carrot powder, while particles subjected to ultrasound showed the primary skeleton of cells that had lost the ambient cell walls through SEM. According to Ivetic et al. [55], the surface structure of sugar beet shreds without ultrasonic pretreatment had no obvious change, and their surfaces were uniform and almost complete. However, most of the cell walls of sugar beet shreds exposed to ultrasonic pretreatment collapsed, the surface structures were destroyed, the internal structures were exposed, and the specific surface area increased. Moreover, compared with compact microstructure of native rice proteins, the protein structure became incompact and more disordered; meanwhile, there were microparticle cracks appearing on the protein surface, which increased the surface area of the protein molecules [56]. Further, compared with the network structure of untreated zein, the surface of the structure under ultrasonic pretreatment underwent great changes. Unlike the original smooth surface of zein, the surface roughness after ultrasonic pretreatment was increased and accompanied by many micro-pores [57]. Li et al. [58] reported that there were a lot of notches and grooves on the surface of corn starch granules after ultrasonic pretreatment, and the surface morphology changed more with an increase in the ultrasonic intensity, which is in line with the results of Shabana et al. [59]. Bai et al. [60] revealed that ultrasound-treated sea bass muscle fibers were severely fragmented, with their borders becoming blurred and evident porosity being observed across the myofibers. These studies indicate that ultrasonic treatment prior to enzymatic hydrolysis could change the surface structure of substrate molecules, including an increase in the roughness of the molecular surface, the production of pores and grooves, and then an increase in the specific surface area of substrate molecules, the destruction of some tight molecular structures, and the relaxation of substrate molecular structures. Thus, enzymes could combine with active groups of substrates more easily in the subsequent enzymatic hydrolysis process to improve the efficiency of enzymolysis [56].

Ultrasonic pretreatment can also peel off targets bound to other substances or disperse aggregated substrates (Figure 1), thus enlarging the contact area with enzymes and promoting enzymatic hydrolysis. Additionally, ultrasonic pretreatment also affected the size of substrate molecules (Figure 1). Ding et al. [61] found that the diameter of grape seed protein particles was much smaller and more uniform after ultrasonic pretreatment (UP) than in the control group. Zhou et al. reported similar results. Atomic force microscopy (AFM) showed that the diameter of protein particles in corn gluten flour was greatly reduced, and its distribution became more uniform after ultrasonic pretreatment. These structural and morphological changes caused by ultrasonic pretreatment could promote substrate and enzyme binding better, thereby improving hydrolysis [62]. However, compared with the control group without ultrasonic pretreatment, NaCas protein pretreated with ultrasound had a less folded surface morphology and larger diameter particles, and its diameter increased from 127 nm to 460 nm, a significant increase of 262.21%. Zhang et al. [63] also found that the diameter of protein particles pretreated by counter-flow ultrasound increased significantly from 275.57 nm to 394.08 nm [64]. These differences may be due to factors such as the ultrasonic form, ultrasonic intensity, and so on, which are discussed below.

The surface hydrophobicity of the substrate has a great influence on the reaction. The surface hydrophobicity, secondary conformation, and surface morphology of proteins changed, suggesting that ultrasonic pretreatment could change the structure of proteins, such as loosening the structure of proteins and exposing hydrophobic amino acids [56,65]. Ultrasound has effects on the tertiary structure of substrates. Zhang et al. evaluated changes in wheat gluten (WG) conformation by studying its surface hydrophobicity (H0). They found that ultrasonic pretreatment significantly increased the H0 of WG over that of the control, which is similar to previous research on the H0 of corn gluten meal, whey protein concentrate, black bean protein isolate, and wheat germ protein [66].

#### 2.1.2. Effect of Ultrasonic Pretreatment on the Structure of Substrates

Ultrasonic pretreatment can change the secondary structure of substrates. Circular dichroism (CD) and Fourier transform infrared spectrum (FTIR) have been frequently used to study the secondary structure of molecules. The mechanical effect of the pretreatment may help to change the secondary structure, as previously reported (Table 1). According to Zhang and Ma et al. [63], the intermolecular β-sheet of wheat gluten protein aggregation and random coil increased by 40.44% and 28.61% compared with the control, while the intramolecular β-sheet, α-helix, and β-turn all decreased. Wang et al. [67] reported that the sweeping frequency and pulsed ultrasound pretreatment increased the β-sheet by 12.4%, while it decreased the α-helix and β-turn by 26.9% and 4.6%, which is consistent with their later report [68,69]. Yang and Li et al. [56] researched the structure characterization of rice protein pretreated with two working ultrasound modes, and the results showed slight reductions in the α-helix and β-turn contents of rice protein and enhancements in the β-sheet and random coil. As the most closed structure in protein molecules, the α-helix is related to the tight structure of the protein. The increase in the coil content may confirm that ultrasound broke down the intrinsic structure of protein molecules, rendering them disordered [70]. Wang et al. [65,71] came to the same conclusion by using various modes of ultrasound. Reducing the α-helix content implies that the tight molecular structure became looser and protein molecules became more stretched. Moreover, the high random coil content meant a much softer and more flexible structure, thus promoting the contact of enzymes with active sites [64,72]. These changes in the secondary structure content might be attributed to the fact that the free radical, shear forces, shock waves, and turbulence induced by ultrasound have disrupted the interactions between the local sequences of amino acids and between the different parts of the protein molecule, and the exposure of hydrophobic groups during the enzymatic hydrolysis of proteins [69,72]. Additionally, Ren and Ma et al. [73] found that sweeping frequency ultrasound treatment on zein resulted in an increase in the α-helix content by 3.4% and increases in the β-sheet, β-turn, and random coil contents by 24.4%, as measured by CD spectroscopy. Zhang and Xu et al. [74] also confirmed that ultrasonic pretreatment weakened the α-helix structure and led to changes in the secondary structure of zeins. Wali and Ma et al. [75] researched the structure characterization of rapeseed protein by multi-frequency ultrasound prior to enzymolysis. They found that the random coil content of ultrasound-treated rapeseed protein had a significant reduction, as it was reduced by 20.40% over the control. These contradictory results might be related to the differences in the native protein and sonication conditions, mainly including the ultrasonic frequency, ultrasonic power density, and ultrasonic mode [56,69].

The increase was attributed to the effect of ultrasonic pretreatment. Ultrasound changed the structure of WG, making hydrophobic groups and regions which were buried in the interior of the protein molecule exposed. Furthermore, Ding et al. reported that the content of total hydrolyzed amino acids, especially the hydrophobic amino acids of hydrolysate and digestion products, increased significantly compared with the control [61]. Additionally, Jin and Ma et al. [72] proved that the H0 of zein increased significantly under sequential dual-frequency ultrasound pretreatment, compared with the control, which increased from 555.9 to 619.4. This observation corroborated the results of Zhang and Ma et al. [80], which showed that the H0 of wheat gluten protein under sequential dual-frequency ultrasound was 2.024-fold higher compared to the untreated sample. When measured by emission fluorescence, the fluorescence peak (450–550 nm) intensity of rice protein under ultrasonic pretreatment increased greatly compared with the control [69]. This increase indicates that intramolecular hydrophobic groups are exposed to molecular surfaces, making it easier for enzymes to combine with the substrate and produce greater reactivity, thus promoting highly efficient enzymatic hydrolysis [81].

Abdualrahman reported that ultrasound shifted the maximum wavelength of the emission fluorescence of the untreated NaCas protein (342.7 nm) to a higher value (345.1 nm), coupled with an increase in the fluorescence intensity. Furthermore, this red shift indicates an increase in the polarity of the tryptophan due to molecular unfolding [64], which promotes enzymolysis of the NaCas protein. This result was in agreement with Li et al. The fluorescence peak (450–550 nm) intensity of pretreated rice protein increased greatly compared to that of the control. The author attributed this phenomenon to the effect of energy-gathered ultrasound as ultrasound could destroy partial hydrophobic interactions of rice protein molecules. Ultrasound induced more molecular unfolding of protein, which caused hydrophobic groups to be exposed outside of the protein molecule [69]. Additionally, Li et al. found that UP decreased the degree of order (DO), double helix (DD), and infrared crystal index (N-O’KI) spectra of corn starch, and the infrared crystallinity index of starch decreased from 1.605 to 1.374. This indicated that the ultrasonic action destroyed the crystallization zone and ordered structures of the starch, which caused a decrease in the partial absorption peak intensity and an improvement in the starch reactivity, providing more opportunities for enzyme-hydrolyzed starch [58].

#### 2.1.3. Effect of Ultrasonic Pretreatment on Thermodynamic and Dynamic Parameters

##### Effect of Ultrasonic Pretreatment on Dynamic Parameters

Enzymolysis reaction kinetics study is highly complex. Generally, it can be described by the Michaelis–Menten kinetic model. The Michaelis–Menten constant (K_M_) value is an important parameter in enzymatic reaction kinetics, which is independent of the substrate concentration. K_M_ represents the affinity between substrates and enzymes during enzymolysis, and the decrease in the K_M_ may result from the high binding frequency between enzymes and the pretreated substrate. Another important parameter in reaction kinetics is the V_max_, which denotes the maximum reaction rate. Ultrasonic pretreatment can influence the kinetics of an enzymatic reaction and further enhance the efficiency of enzymatic hydrolysis by changing the structures of substrates.

Ayim et al. reported that the kinetic parameters of enzymatic hydrolysis changed after single-frequency counter-current ultrasound pretreatment. Compared with traditional enzymatic hydrolysis, the rate constant K_M_ of ultrasound-pretreated tea residue protein (UTRP) exposed to enzymatic hydrolysis decreased by 32.7% [82]. Similar findings were reported by Zhang et al., where the K_M_ for ultrasound-pretreated pectin decreased from 3.28 mg/mL to 2.99 mg/mL, compared with a non-pretreated sample [80]. The decrease in the K_M_ value implies an increase in the affinity between substrates and enzymes. This change might be attributed to the significant role of ultrasonic pretreatment in the enzymatic reaction. On the one hand, ultrasonic pretreatment weakened the hydrogen bond, Van der Waals forces, and hydrophobic interactions, thus loosening the protein structure and making it easier to undergo enzymatic hydrolysis. On the other hand, ultrasonic pretreatment could change the surface morphology of the substrate and increase the contact area between the enzyme and substrate [82,83]. Further, Ma et al. found that ultrasonic pretreatment had a positive effect on the kinetics of pectin enzymolysis. After ultrasonic pretreatment, the K_M_ value of the enzymatic kinetic parameters decreased, while the V_max_ value increased by 29.41%. The increase in the V_max_ meant that the enzymatic hydrolysis efficiency of pectin and pectinase improved [84]. Similar results were reported by Zhou et al. [62]. It was proven that the K_M_ value decreased by 10.62% and the V_max_ increased by 21.48% during the degradation of carboxymethylcellulose pretreated by ultrasound. Interestingly, with regard to V_max_, contrary results were reported by some studies, where V_max_ showed a slight downward trend [67,83,85]. K_M_ is fundamental to the kinetics of the hydrolysis reaction, which is inversely proportional to the reaction rate: the lower the K_M_ value, the faster the reaction rate [86]. Ariana de Souza Soares et al. [51] came to a different conclusion, finding that ultrasound as a pretreatment method did not promote sucrose hydrolysis but slightly reduced the invertase activity. Sonication increased Vmax (increased of 23%) and maintained a constant Km, showing that the ultrasound sped up the mass transfer but did not affect the enzyme during the reaction. In addition, a constant (k) rate of reaction is an indispensable factor in the kinetics of chemical reactions. Wang et al. [65] reported single-frequency counter-current S-type and multi-frequency counter-current S-type ultrasonic pretreatments on defatted corn germ protein. From the results, compared to traditional enzymatic hydrolysis, K increased.

Additionally, the association constant (K_A_), as the average value of the apparent breakdown rate constant, was used to show the degree of the binding frequency between substrates and enzymes [87]. Some reports are summarized in Table 2. Compared with traditional enzymatic hydrolysis, the K_A_ value for pretreatment with dual-frequency ultrasound increased by 1.96%. The authors attributed this result to better mixing of the substrate and enzyme after ultrasonic pretreatment [44]. Similar results were reported by others [62,64]. However, some researchers also found that ultrasonic pretreatment reduced the enzymatic parameter K_A_ [88]. The reason for this opposite result may be the difference in the ultrasound conditions and mode or ultrasonic equipment. Generally, ultrasonic pretreatment has profitable effects on enzymatic hydrolysis, such as increasing the affinity between enzymes and substrates, enhancing the frequency of binding between enzymes and substrates, and increasing the maximum rate of enzymatic hydrolysis, thereby enhancing the enzymatic hydrolysis efficiency [89].

##### Effect of Ultrasonic Pretreatment on Thermodynamic Parameters

The thermodynamic parameters of the enzymolysis reaction mainly include the activation energy (E_a_) and the changes in the enthalpy (ΔH), entropy (ΔS), and Gibbs free energy (ΔG). The E_a_ is the minimum energy required to convert a stable molecule into a reactive molecule, and it reflects the speed and rate of the chemical reaction. It has been reported that most of the reactions require activation energies ranging from 40 to 400 kJ/mol. When the E_a_ is less than 40 kJ/mol, the reaction proceeds rapidly [81]. The ΔH refers to the energy required to convert substrates into products. A lower ΔH means a lower energy cost [43]. The ΔS represents a change in the degree of local disorder between the transition state and the ground state; moreover, the ΔG is closely related to the ΔH and ΔS.

The effects of ultrasonic pretreatment on the thermodynamic parameters of the enzymolysis reaction have been reported by many researchers, as shown in Table 3. Ren et al. reported that the E_a_, ΔH, ΔS, and ΔG of hydrolysis with sweeping frequency ultrasonic pretreatment decreased by 19.5%, 20.63%, 6.16%, and 7.02%, respectively [43]. The result was in line with Dabbour et al. Dabbour et al. found that the thermodynamic parameters of enzymatic hydrolysis of sunflower proteins changed after treatment with dual-frequency ultrasound (DFU). Regarding Arrhenius kinetics, DFU reduced the E_a_, enthalpy, and entropy by 24.28%, 26.13%, and 9.10%, respectively [44]. Similar results were obtained by other researchers [65,76,78,79]. The decrease in E_a_ implies that ultrasonic pretreatment reduced the energy limit of the enzymatic reaction, making it easier to carry out the reaction and thus improving the reaction efficiency. The decrease in ΔH indicates that ultrasonic pretreatment could change the protein structure by destroying the hydrophobic interaction, stabilizing the relationship between enzymes and proteins, adapting to the oxidation of amino acid residues, and converting enzymes and substrate complexes into active states [87]. The orderly distribution of enzymes and substrates during the reaction process after ultrasonic pretreatment might be responsible for the decrease in ΔS, which was the result of enhanced affinity between the substrate and enzyme. Moreover, the negative ΔS value indicates that the entropy decreases during enzymatic hydrolysis [44]. Generally, ultrasonic pretreatment changes the structure of substrates, such as hydrophobic groups or region exposure, and surface morphology changes, thus promoting enzymolysis.

### 2.2. Effect of the Ultrasonic Pretreatment Conditions on Enzymolysis

Ultrasonic-pretreatment-assisted enzymatic hydrolysis is affected by ultrasound parameters. Several ultrasound parameters, such as the ultrasound frequency, ultrasound intensity, and duration of irradiation, determined any potentially damaging effects on the biological molecules. Hence, a change in the ultrasound parameters further affects enzymatic hydrolysis.

Yang et al. studied the effects of ultrasonic pretreatment on rice protein at different frequencies and in different working modes, including single-frequency ultrasound, dual-frequency ultrasound, and triple-frequency ultrasound, using the hydrolysis degree (DH) and angiotensin-I-converting enzyme (ACE) inhibitory activity as indicators. Although ultrasonic pretreatment did not significantly improve the DH, ACE inhibitory activity was significantly different under different ultrasound modes and frequencies. Compared with other ultrasound frequencies and modes, sequential triple-frequency ultrasound (20/35/50 kHz) showed the highest ACE inhibitory activity. Compared with other frequencies of single-frequency ultrasound, 20 kHz single-frequency ultrasonic pretreatment showed excellent ACE inhibitory activity [56]. Inversely, Wang et al. [65] investigated the influences of different frequency modes of a low-power density ultrasound (LPDU) on the enzymolysis efficacy and structural property of corn gluten meal (CGM).The results show that sequential DFU with 20/40 kHz was the most efficient. Thus, the more combined ultrasonic frequency settings were not better. Interestingly, the sequential operating modes were more conducive to improving the relative enzymolysis efficiency of CGM during ultrasonication pretreatment, instead of using simultaneous operating modes. In addition, the difference in enzymatic hydrolysis under different ultrasound frequencies and modes may be attributed to the difference in the ultrasonic effect. Ding and Wang et al. [70] studied the effects of ultrasonic pretreatment with three modes of cyclic-sweeping frequency, random-sweeping frequency, and fixed frequency on the secondary structure of rice protein, and the experiment showed that the random-sweeping frequency ultrasound had the most significant effect on the structure of rice protein. Li et al. reported that different ultrasound conditions caused different starch hydrolysis effects. Suitable ultrasonic pretreatment conditions could create better conditions for maize starch enzymatic hydrolysis [58]. Under three ultrasound conditions (U_1_: 40 min, 420 W, 50 °C; U_2_: 30 min, 480 W, 40 °C; U_3_: 20 min, 540 W, 60 °C), the pasting properties and molecular weight distribution of corn starch changed. In particular, compared with U_1_ and U_3_, the values of PV, TV, BV, and FV and the molecular weight of U_2_ changed more. This phenomenon might be attributed to the destruction of the starch crystallization zone by ultrasound, the reduction of the starch degree of polymerization, the hydrolysis of the starch long chain, and the exposure of a large number of non-reducing ends, which provide more binding opportunities for liquefied enzymes to hydrolyze starch [54]. Several similar reports are summarized in Table 4.

Several researchers have also concluded that temperature is an important parameter [51,65]. Ge and Tong et al. [94] showed that the ultrasonic frequency and temperature have significant effects on the extraction of silkworm pupa protein. With the increase in temperature, the solubility of the silkworm pupa protein in the solvent increases, increasing protein production. Cheng et al. [88] came to the conclusion that ultrasound decreases the inactivation temperature of glucoamylase. As the temperature goes up, the combination of thermal inactivation and ultrasonic inactivation leads to a reduction in glucoamylase activity. Therefore, ultrasound results in more serious inactivation of glucoamylase at high temperatures. Wang et al. [71] stated that when cavitation bubbles collapse at a higher temperature, the vapor pressure of the solution increases to accompany the maximum pressure value generated, thence causing a reduction in the cavitation action.

In addition, Wang et al. [71] found that the ultrasonic interval ratio affects the formation of cavitation bubbles, and if the time is too long, the bubbles may burst before formation. A long ultrasonic time will lead to the aggregation of protein molecules, which reduces the interaction between enzymes and substrates.

## 3. Ultrasound–Enzyme Synergy (UES) in Protein Hydrolysis

Recently, many studies have shown that ultrasound can enhance enzyme activity to some extent, suggesting that different enzymes have different levels of tolerance and sensitivity to ultrasound [50,95]. UES affects enzymes and substrates and promotes their binding (Figure 2C). This is discussed later.

### 3.1. Mechanism of UES

#### 3.1.1. Effect of UES on the Enzyme Structure

The structure of enzymes plays an important role in the catalytic efficiency and stability of enzymes. Because the chemical nature of enzymes includes the primary, secondary, tertiary, and quaternary structures, as in the case of proteins [2], the structure of enzymes is easily affected by physical and chemical factors, such as ultrasound. Several studies have reported that ultrasound can cause changes in the surface morphology (Figure 2B) and secondary and tertiary structures of enzymes.

Wang et al. reported that the surface morphology of immobilized enzymes treated with dual-frequency ultrasound changed from smooth to rough and became cracked compared with the control group (without ultrasonic treatment). In addition, enzymatic hydrolysis at the same time yielded a large number of polypeptides, which indicated an improvement in the enzymatic hydrolysis efficiency [96]. This is similar to their earlier experimental results. Wang et al. found that mild ultrasound could loosen the cellulase structure and increase the surface area of cellulase [40]. These changes may be due to the strong shear force produced by ultrasonic cavitation, which affects the surface morphology of the enzyme. In the subsequent enzymatic hydrolysis process, the surface morphology of the enzyme was rough and the specific surface area was large, which makes it easier for substrates to combine with the enzyme, thus greatly improving the enzyme activity [40]. Others reported that ultrasound may expose the active site of the enzyme and increase the affinity of enzymes with substrates, thereby increasing the rate of enzymatic hydrolysis [96,97,98].

The primary structure of enzymes is composed of amino acids linked with peptide bonds. These polypeptide chains tend to form secondary structures, including α-helices, β-sheets, β-turns, and random coils via hydrogen bonding (Table 5). Such secondary structures tuck over three-dimensionally to become the subunit establishing the tertiary structures through hydrophobic interactions, while the quaternary structure comprises those subunits associated via Van der Waals attractive forces. Modification of the structure of an enzyme will alter its functionality, stability, and residual activity. Thus, inactivating the enzyme requires the alteration of its structure by affecting the molecular interactions among the amino acids [2]. Generally, an enzyme is a kind of biocatalyst whose catalytic ability mainly depends on its active center [99]. Ultrasound mainly affects the catalytic reaction by affecting the activity of enzymes. Earlier studies have found that ultrasound can reduce the activity of enzymes and even inactivate them [100]. However, recent studies have shown that the appropriate ultrasound treatment not only does not reduce the activity of the enzyme but also promotes the activity of the enzyme [3,101,102]. This may be due to the effect of ultrasound on the structure of enzymes, which changes the enzymes into more catalytic structures. With ultrasonic pretreatment at a 90% amplitude for 30 min, edible bird’s nest hydrolysate (EBNH) increased its anti-lipoxygenase and alpha-amylase inhibitory activities by 100% and 43%, respectively, and ultrasonic pretreatment is recommended as an effective green upstream process to enhance the anti-inflammatory and hypoglycemic activities of EBNH [103]. Therefore, the enzymatic activity can be improved by the appropriate ultrasound treatment, and then, the rate and yield of the enzymatic reaction can be increased.

The contents of the α-helix (23.4%) and β-sheet (25.0%) of cellulase treated by ultrasound were lower than those of untreated samples (26.2%, 26.6%). In addition, the enzymatic activity increased by 18.17% during ultrasound-assisted enzymatic hydrolysis [40]. Feng et al. [109] reported that the secondary conformation of papain treated by ultrasound was quite different from that of untreated papain. The main difference lay in the contents of the α-helix, β-sheet, and random coil. The contents of the α-helix and random coil in papain treated by ultrasound were higher than those of untreated samples, while the content of the β-sheet was lower than that of untreated samples. This is similar to the results of Xu et al. [110], who showed that the α-helix content of polyphenol oxidase dropped to 16.9% and the β-sheet content dropped to 25.9% at 22/40 kHz (residual enzyme activity 5.84%). Interestingly, Ma et al. reported a slightly different result from theirs, showing that the content of the α-helix increased by 5.2% and that of the random coin decreased by 13.6% in alcalase protein; however, its activity still showed an improvement [99]. Although the changes in the secondary conformation in these studies were different, they all improved the enzyme activity to a certain extent. The mechanism of changes in specific secondary conformations of enzymes by ultrasound may not be clear. However, it could be suggested that conformational changes in the secondary structure enhance the formation rate of the enzyme substrate complex as well as the release rate of the product from the enzyme [111]. Namely, these secondary conformation changes caused by ultrasonic treatment may make enzyme molecules stretch and become more flexible, thus making it easier for substrates to combine with the active center of the enzyme, thereby improving enzymatic activity [99].

The fluorescence spectrum is considered to be an effective method for determining the tertiary structural and conformational changes of proteins, because the intrinsic fluorescence of aromatic amino acid residues, mainly including tryptophan, tyrosine, and phenylalanine residues, is sensitive to the polarity of the microenvironment during the transition process. Nadar et al. found that the intrinsic tryptophan fluorescence intensities of pepsin and amylase treated with 30 min ultrasound were higher than those of untreated samples, and enzymatic activity was significantly enhanced [112]. Nadar et al. obtained similar results [112]. The intrinsic tryptophan fluorescence intensity of lipase with ultrasonic treatment increased compared to that of unsonicated lipase, which indicates that ultrasonic treatment changed the number of tryptophan residues on the surface of lipase. Guo et al. [113] used the ANS probe method to study the surface hydrophobicity of horseradish peroxidase pretreated with ultrasound. The experiment found that the hydrophobic groups inside horseradish peroxidase were exposed to a hydrophilic environment, indicating molecular unfolding and a loss of the tertiary structure. In addition, there were other explanations for the changes in the fluorescence intensity of enzymes under ultrasound. This change was mainly attributed to the hydrophobic interaction between proteins or the exposure of hydrophobic groups and regions within the molecules [109]. These results agree with those of Yang et al. and Li et al. [56,69]. Due to the effect of ultrasound, the unfolding of protein molecules and hydrophobic groups, which were buried in the molecule at the beginning, was exposed. Moreover, ultrasound affects enzyme stability and activity; some enzymes undergo intermolecular interactions in the polypeptide chains by breaking [100]. Under the action of ultrasonic cavitation, the hydrophobic surface of the protein increases, and the complex structure of protein is destroyed with the stretching of protein molecules [62]. This phenomenon may also result from the partial denaturation of the enzyme caused by the strong shear forces and micro-jet produced by the collapse of cavitation bubbles under the action of ultrasound.

#### 3.1.2. Effect of UES on the Substrate Structure

If ultrasound is applied to the whole system when enzymatic hydrolysis occurs, the thermal effect of ultrasound can increase the temperature of the reaction system and promote the reaction to a certain extent. In addition, when the ultrasound vibrates, the medium particles enter the vibration state at a very high speed and accelerate through energy transfer, which can also increase the contact opportunities between the enzyme and the substrate to a certain extent and then change the process of enzymatic hydrolysis. In the ultrasound-assisted enzymatic (UAE) process, the changes in substrates were similar to those of ultrasound-assisted pretreatment alone. The changes can be summarized by two points. On one hand, ultrasound changes the surface morphology of the substrate molecule or the size of the substrate particles, thus expanding the contact area with the enzyme, thereby increasing the enzymatic hydrolysis efficiency (Figure 2A). On the other hand, ultrasound modifies the secondary and tertiary structures of the substrate molecule to a certain extent, making it easier to bind to enzymes. Yang et al. reported that sisal waste was relatively compact, complete, and smooth before extraction. UAE resulted in the collapse of the microstructure and relatively rough morphology for the sisal material [41]. The result is similar to that of Heidari et al [114]. The morphology of initial peanut seed powder was regular, complete, and smooth, while it showed disintegration and porosity after UAE [114] treatment. Wang et al. [100] researched the mechanism of ultrasound-accelerated enzymatic hydrolysis of starch. The ultrasound-induced chain breakage resulted in a molecular weight decrease. The degradation process could be attributed to cavitation, which creates high temperatures, high pressures, and shear forces that can break the starch chains. Moreover, Zhang et al. [115] showed the preparation of microkeratome and nanokeratome by ultrasound-assisted enzymatic hydrolysis. They believe that ultrasonication plays important roles in extraction. First, the shear stress supported by ultrasound allows keratin to be effectively separated before the matrix is completely dissolved. Second, ultrasound irradiation can promote the reactions for the extraction of micro- and nanokeratin, including the enzymatic hydrolysis of proteins, disulfide bond breakage by reductants and the disruption of hydrogen bonds. Ma et al. reported that ultrasonic treatment could change the structure of β-lactoglobulin. Compared with the hydrolysis of β-lactoglobulin without ultrasonic treatment, β-lactoglobulin treated by ultrasound and enzymes showed an improvement in the α-helix and β-sheet structures [116]. Wang et al. researched the mechanism of ultrasound-accelerated enzymatic hydrolysis of starch [100]. They found that ultrasound can effectively damage the starch cluster structure, destroy the starch chains, and reduce the molecular weight of starch. Ultrasound increased the free mobile starch fragment and exposed more sites to react with enzymes.

### 3.2. Key Parameters in UES

The UAE process is affected by many parameters. The frequency of ultrasound determines the cavitation effect of ultrasound [47]. In addition, when the enzyme is subjected to the optimal ultrasonic frequency, its activity increases [48]. Wang et al. [96] studied the effects of the ultrasonic power intensity, enzyme–substrate ratio, substrate concentration, and ultrasonic time (enzymatic hydrolysis time) on the UAE process in detail. The activity of enzymes and the degree of hydrolysis (DH) of substrates were used as indicators. The DH of substrates was affected by the ultrasonic power density. Having the appropriate ultrasonic power density is not only beneficial to hydrolysis but also enhances enzyme activity. Ma et al. [117] investigated the effects of the ultrasonic frequency, ultrasonic time, and temperature on lignin removal and sugar production through the enzymatic hydrolysis of corn cob and significantly improved the cellulose saccharification rate under optimal treatment conditions. Zhu et al. [118] studied the effects of the ultrasonic time, ultrasonic power, enzyme addition amount, enzymatic hydrolysis time, pH value, and temperature changes on the enzymatic hydrolysis effect of ultrasonic pretreatment. The results showed that the content of soluble solids and the antioxidant activity of the optimized edible fungi by-product hydrolysate significantly increased. Yun et al. [119] found that the optimal extraction conditions for the ultrasonic-assisted enzyme extraction of *S. baicalensis* root polysaccharide were as follows: a cellulase concentration of 165.6 U/mL, a temperature of 57.3 °C, a liquid–solid ratio of 44.8 mL/g, a time of 50 min, and an ultrasonic power of 225 W. The obtaining rate of the *S. baicalensis* root polysaccharide was as high as 12.27%. Similar results were reported by others [44,57]. The reasons for this beneficial phenomenon could be summarized as follows: (1) At a low power density, the thermal effect of ultrasound increases the temperature of the solution, resulting in decreases in the substrate viscosity coefficient and surface tension coefficient, and then the cavitation threshold decreased, which made it easier to produce cavitation bubbles [13]. (2) Having a suitable ultrasonic power density changed the conformation of the enzyme, which resulted in the binding sites of the enzyme being more suitable for binding to the substrate [96].

The enzyme–substrate ratio and the substrate concentration also affected the UAE process. Wang et al. [96] found that the DH and enzymatic activity first increased and then decreased with the increase in the enzyme–substrate ratio. Generally, the DH decreased by increasing the substrate concentration, which showed the opposite tendency to enzyme activity. Zhang et al. reported similar results, showing that an increase in the substrate concentration, the DH of wheat gluten decreased, while its ACE inhibitory activity increased significantly [66]. Interestingly, Ding et al. reported the opposite result. With an increase in the substrate concentration, the DH of grape seed protein increased [61]. Although this change was small, it cannot be ignored. These results could be attributed to differences in the amounts of enzymes and substrates. The DH depended on the number of enzymes and the concentration of substrates. When the enzyme was fully combined with the substrate, the DH increased. However, when the enzyme was excessive, the supply of the substrate was insufficient, and the interactions between highly concentrated enzymes might lead to enzymatic decomposition. In addition, the excessive increase in the substrate concentration led to a decrease in enzyme–substrate binding, which reduced the rate of enzymatic hydrolysis [67,96].

The UAE process has been reported in many studies to promote the whole enzymatic hydrolysis process. Thereinto, the ultrasonic time is an important parameter of enzymatic hydrolysis. With an increase in the ultrasonic time, the DH of substrates, such as rapeseed protein, soy sauce residue, and red seaweed increased [116,120,121]. However, with an increase in the ultrasonic time, the enzyme activity generally decreased [121]. Although the enzyme activity would not decrease under ideal conditions, with the extension of the ultrasonic time, the extreme temperature and pressure under ultrasound, as well as the production of free hydroxyl radicals and hydrogen radicals, led to a decrease in, or even inactivation of, the enzyme activity [122]. Therefore, the appropriate ultrasonic time is the key to promoting enzymatic hydrolysis. In addition, the ultrasonic temperature can affect the reaction. Hao et al. reported that as the ultrasonic power increases, the conversion rate required to decrease the activity of lipase is affected to a certain extent. However, the decreasing trend in the conversion rate becomes stable gradually [122].

### 3.3. Application of Ultrasound-Assisted Enzymatic Hydrolysis in Food Processing

Compared with traditional enzymatic hydrolysis, which has several disadvantages, such as a low contact frequency with substrates, a long enzymolysis time, low enzyme activity, and the aggregation of substrates, ultrasound-assisted enzymolysis can effectively solve these problems. As a new technology, many papers have reported applications of ultrasound-assisted enzymatic hydrolysis technology, such as for biodegradation, biological fermentation, and chemical synthesis. Due to its high efficiency, ultrasound-assisted enzymolysis technology has also been widely applied in food processing.

In food processing, the applications of ultrasound-assisted enzymatic hydrolysis technology can be divided into extraction and degradation. The disadvantages of traditional extraction technology related to the ecological environment led to the emergence of this new extraction technology [3]. Ultrasound-assisted enzymatic extraction could be used to extract several compounds, including phenols, protein, pectin, and peptides [123,124,125,126]. Nag et al. reported that ultrasound-assisted aqueous enzymatic extraction (UAEE) can be used for the recovery of polyphenols from pomegranate peels, which indicates that UAEE is a promising method for obtaining polyphenols from waste and can be utilized to extract important biomolecules from agricultural and food waste without using chemical solvents [127]. Wang et al. used ultrasonic pretreatment and UAEE to improve the oil recovery of gardenia fruit. Compared with the untreated powder, the ultrasonic-pretreated powder (480 W, 30 min) had a higher oil yield. Ultrasonic pretreatment affected the composition, rheology, and surface morphology of the powder, made the sample more susceptible to enzyme attack, and finally increased the oil extraction rate [128]. In addition, ultrasound-assisted enzymatic hydrolysis can also be used in hydrolysis, especially for protein hydrolysis, such as for obtaining antioxidant peptides from corn protein and ACE inhibitory peptides from zein and wheat gluten [43,44,129]. Similarly, Wali et al. obtained ACE inhibitory peptides through the ultrasound-assisted enzymatic hydrolysis of rapeseed protein. Compared with the control group, ultrasound-assisted enzymatic hydrolysis increased the ACE inhibitory activity of hydrolysates [75]. The ultrasonic-assisted method not only significantly improved the extraction yield of Glycyrrhiza uralensis seed protein (GSP-U) but also enhanced its functional properties and biological activities. After hydrolysis, the enzymatic hydrolysates also showed more functional properties and antioxidant activity [130]. Additionally, ultrasound-assisted enzymatic hydrolysis can be roughly divided into the following applications according to its purpose, as shown in Table 6.

## 4. Devices for Ultrasound-Assisted Enzymatic Hydrolysis

In the food processing industry, ultrasound is widely used due to its high efficiency. Ultrasonic devices consist of two parts: an ultrasonic generator and an ultrasonic transducer. The basic principle of an ultrasonic device is that the high-frequency electric oscillation produced by the ultrasonic generator is applied to the ultrasonic transducer. The electric oscillation signal causes a change in the electric field or magnetic field in the original part of the electrical energy storage in the transducer. Through some effect, it generates a driving force on the mechanical vibration of the transducer, thus promoting the vibration of the medium in contact with the mechanical vibration system of the transducer and radiating into the medium.

Traditional high-power ultrasonic devices are mainly divided into ultrasonic bath devices and ultrasonic horn devices (Figure 3). Compared with the ultrasonic probe, the ultrasonic bath, which is the most common method of ultrasonic treatment, has a lower cost and larger sample processing capacity; however, its low repeatability and low power are its main shortcomings [146]. Probe-type ultrasound can reduce the resonance impedance and improve the electro-acoustic conversion efficiency through a horn. It can transmit ultrasound directly into the sample medium, the energy loss is lower, and the intensity of the ultrasound is higher than that of the ultrasonic bath. The probe-type ultrasonic device is usually the first choice for sample pretreatment. However, due to the differences in the practical application and the volume of the sample to be measured, it is necessary to select the appropriate length, diameter, and tip geometry for the probe design [146]. Moreover, probe-type ultrasonic equipment is not easy to amplify and is difficult to apply in industrial production. As the temperature of medium increases due to the thermal effect of ultrasound, the traditional ultrasonic equipment is usually improved to control the reaction temperature (Figure 3). Based on the shortcomings of discontinuous materials, the research team of Ma improved the traditional ultrasound and developed a new counter-current ultrasound device (Figure 3). Compared with the single ultrasonic bath and ultrasonic probe, counter-current ultrasound has both advantages; moreover, counter-current ultrasound is more uniform.

According to the type of working frequency, ultrasound can be divided into single-frequency ultrasound, dual-frequency ultrasound, and multi-frequency ultrasound. On the basis of single-frequency ultrasonic equipment, by installing ultrasonic transducers with two or more different frequencies, the ultrasonic transducers of different frequencies are driven by several ultrasonic generators, so as to achieve the effect of having multiple frequencies. It was reported that the energy efficiency of dual-frequency ultrasound was more than twice that of single-frequency ultrasound [147]. Many studies have confirmed that the effects of dual-frequency ultrasound or multi-frequency ultrasound are better than those of single-frequency ultrasound [147,148,149]. In particular, the research team of Ma developed a new type of multi-frequency ultrasound equipment, which can work in mono-, dual-, or tri-frequency modes.

According to the mode of transmitting ultrasonic frequency, ultrasound can be divided into two forms: fixed-frequency and sweep-frequency. Compared with the fixed-frequency ultrasound, the frequency of sweep-frequency ultrasound fluctuates around the central frequency for a certain sweep period. Sweep-frequency ultrasound can better excite the resonance frequency matching the natural frequency of the sample solution, and achieve a better treatment effect, such as applications in the pretreatment of corn gluten powder and gluten powder [68,77]. From an acoustical point of view, sweep-frequency ultrasound can produce a sound field, which is more conducive to improving the cavitation effect. The propagation of sweep-frequency ultrasound in the media would cause stronger vibration and high acceleration, which could increase the frequency and speed of molecule movement and increase the penetration force of solvents [150,151].

## 5. Conclusions and Future Perspectives

This review summarizes the mechanism of enzymatic hydrolysis promoted by ultrasound reported in recent years. Ultrasonic pretreatment before enzymatic hydrolysis can effectively change the structure of the substrate, including an increase in the surface roughness of the sample, a reduction in the molecular size, and a loosening of the molecular structure of the sample. These changes make it easier for substrates to bind to enzymes, thus increasing the efficiency of enzymatic hydrolysis. Ultrasonic pretreatment can weaken the hydrogen bonds, Van der Waals forces, and hydrophobic interactions of proteins, thus making the protein structure loose and easier to hydrolyze by enzymes. When ultrasound is applied to the whole enzymatic hydrolysis system, the USE system provides an appropriate environment for the reaction by changing enzymes, substrates, and the reaction between enzymes and substrates, thus promoting enzymatic hydrolysis. In the process of UES, the secondary conformational change induced by ultrasonic treatment may make the enzyme molecule stretch more flexible, change the enzyme to a more catalytic structure, and thus increase the rate of formation of enzyme substrate complexes and the rate of product release from the enzyme. In addition, ultrasound affects the stability and activity of enzymes. Therefore, appropriate ultrasonic treatment can improve the activity of the enzyme and increase the rate and yield of the enzymatic reaction. The kinetic and thermodynamic changes of enzymatic hydrolysis with or after ultrasound treatment further indicate the effectiveness of ultrasound-assisted enzymatic hydrolysis. The UES system changes the kinetic parameters, indicating that UES can improve the efficiency of enzymatic hydrolysis by increasing the affinity between the enzyme and substrate, increasing the binding frequency between the enzyme and substrate, and increasing the maximum rate of enzymatic hydrolysis. The changes in thermodynamic parameters indicate that UES improves the reaction efficiency by reducing the energy limit of the enzymatic reaction, destroying the hydrophobic action and changing the protein structure, and transforming the enzyme and substrate complex into an active state to make the reaction easier to carry out. In the actual food processing process, UES can decompose proteins into small peptides and amino molecules and can also release other special properties and functional active substances, which can be used to improve the food flavor and nutrition. Ultrasound equipment has been greatly improved and can better help UES to improve the efficiency of enzymatic hydrolysis.

By increasing the surface roughness of the substrate, exposing the internal hydrophobic groups, and making the structure disordered and loose, ultrasonic pretreatment increases the contact and binding opportunities between the substrate and the enzyme, thus improving the efficiency of enzymatic hydrolysis. In the future, in the process of ultrasonic pretreatment, the addition of edible denaturants that promote the structural decomposition of the substrate protein and assist ultrasound to further accelerate the loosening of the protein structure can be considered. At the same time, it is possible to reduce the energy use of ultrasound equipment and contribute to the research of more green and environmentally friendly ultrasonic pretreatment enzymatic hydrolysis protein technologies. Ultrasound pretreatment can promote enzyme binding to the substrate. Whether this is due to the exposure of the binding site or other reasons needs to be further studied. In addition, whether the ideal effect is achieved after ultrasonic pretreatment cannot be judged only by the hydrolysis effect. How to quickly, simply, and effectively detect whether the ultrasonic pretreatment effect is up to standard is also worth studying. UES is suitable for hydrolyzing systems with low sensitivity to ultrasound and highly viscous products. Ultrasound changes the conformation of the enzyme and substrate, which leads to an improvement in the enzyme catalytic efficiency. However, the mechanism by which UES improves the efficiency of enzymatic hydrolysis needs further study. In the synergistic process of ultrasonic enzymes, it is necessary to screen out the enzymes that are not easily inactivated by ultrasound and have good tolerance to ultrasound. In addition, it is necessary to optimize the UES process parameters to better improve the enzymatic hydrolysis efficiency and reduce the energy consumption of ultrasound equipment. The current ultrasonic equipment is more efficient in the laboratory. However, determining how to find simple and efficient ultrasonic equipment with a good effect, low cost, and easy amplification production and realizing the amplification production of ultrasonic equipment are also topics that are worth studying in the future.

## Figures and Tables

**Figure 1 foods-12-04027-f001:**
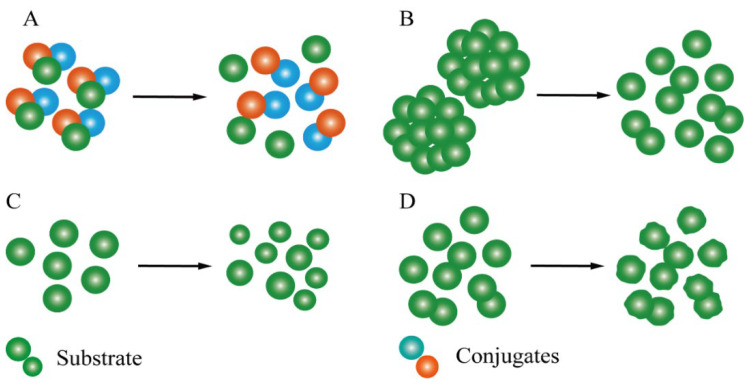
Effect of ultrasonic pretreatment on the substrates: (**A**) dissociation of targets from other substances; (**B**) dispersion of aggregated substrates; (**C**) size reduction from large substrates to small molecules; (**D**) changes in the surface morphology of substrates.

**Figure 2 foods-12-04027-f002:**
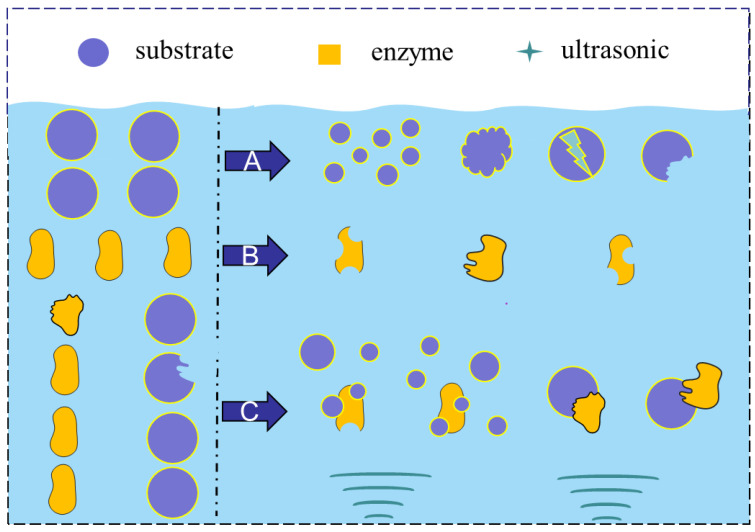
Effects of UES on the substrate structure (**A**), enzyme conformation (**B**), and binding of the substrate and enzyme (**C**).

**Figure 3 foods-12-04027-f003:**
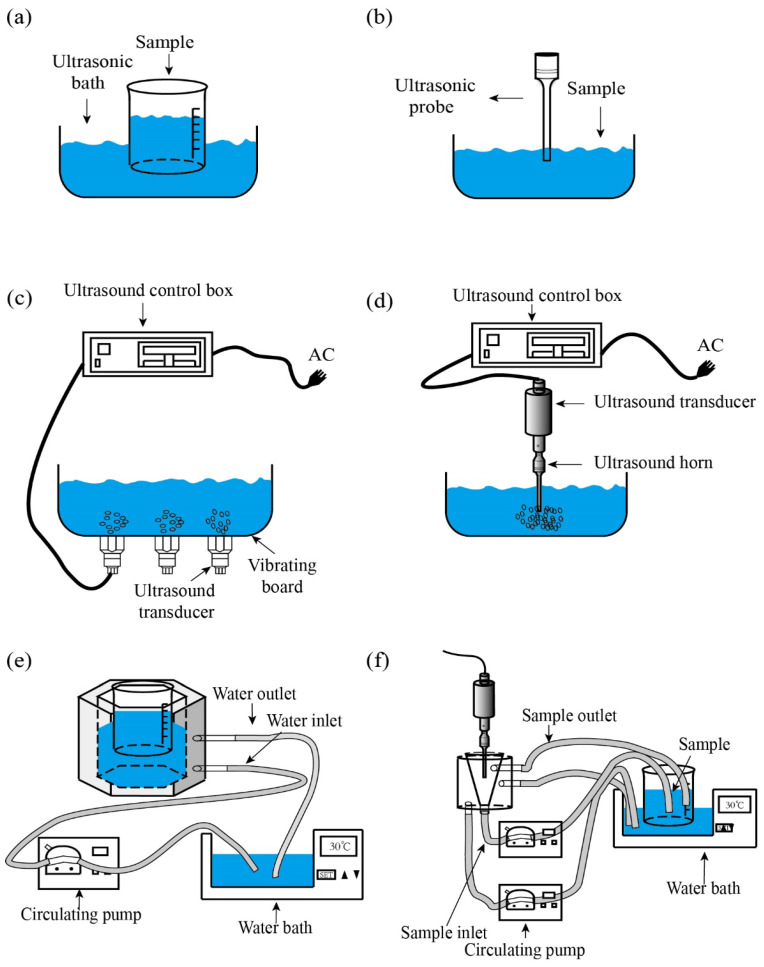
Typical and modified concepts of ultrasonic equipment: (**a**) ultrasonic bath; (**b**) ultrasonic probe; (**c**) ultrasonic bath equipment with frequency and power control; (**d**) probe-type ultrasonic equipment with frequency and power control; (**e**) ultrasonic bath with a temperature control device; (**f**) continuous counter-current sonication with an ultrasound probe and temperature control device.

**Table 1 foods-12-04027-t001:** Effect of ultrasonic pretreatment on the secondary structure of substrates.

Substrate	Type	α-Helix (%)	β-Sheet (%)	β-Turn (%)	Random Coil (%)	Reference
Corn gluten meal	*	4.1	42.1	-	53.8	[62]
	**	3.4 (−17.07%)	42.3 (0.47%)	-	54.2 (0.74%)	
Feather protein	*	14.80	43.43	27.55	14.22	[4]
	**	13.47	42.01	26.30	18.22	
Brewer’s spent grain protein	*	7.41	44.71	31.07	16.82	[76]
	**	13.99 (+6.58%)	23.21 (+21.50%)	45.89 (+14.82%)	16.92 (+0.10%)	
Rice protein	*	18.85	27.76	35.02	18.37	[56]
	**	18.06 (−4%)	28.69 (3%)	33.95 (−3%)	19.30 (5%)	
Sodium caseinate protein	*	51	12	15	22	[64]
	**	46.5 (−8.82%)	14.4 (20%)	15.8 (5.33%)	23.3 (5.91%)	
Rapeseed protein	*	32	32.3	0.5	35.3	[75]
	**	31.6 (1.25%)	40.3 (23.84%)	0 (−100%)	28.1 (−20.40%)	
Wheat gluten	*	47.70	13.90	15.40	23.00	[77]
	**	46.30 (−1.3%)	13.90 (0%)	15.70 (10.3%)	24.00 (1%)	
	*	17.69	5.96	10.31	15.24	[63]
	**	12.96 (−26.74%)	8.37 (40.44%)	9.83 (−4.66%)	19.60 (28.61%)	
Gelatin	*	9.98	44.02	23.24	22.76	[78]
	**	7.18 (−28.06%)	42.82 (−2.73%)	30.75 (32.31%)	19.26 (−15.38%)	
Whey protein	*	46.23	11.25	16.26	26.25	[79]
	**	44.75 (−3.20%)	12.28 (9.16%)	16.60 (2.09%)	26.37 (0.46%)	

* Control (without ultrasound); ** pretreated by ultrasound. Values in parentheses indicate an increase or decrease (with negative sign) in the secondary structure (α-helix, β-sheet, β-turn, and random coil).

**Table 2 foods-12-04027-t002:** Effect of ultrasonic pretreatment on the enzymolysis kinetics of several substrates.

Substrate	Treatment	K_A_ (min^−1^)	K_M_ (g/L)	Reference
Rapeseed protein	*	0.075	13.940	[87]
	**	0.083 (+10.67%)	11.490 (−17.58%)	
Corn gluten meal	*	1.066	499.870	[62]
	**	1.183 (+10.98%)	491.747 (−1.63%)	
Potato protein	*	0.510	13.726	[88]
	**	0.490 (−3.92%)	8.572 (−37.55%)	
Sodium caseinate protein	*	0.754	25.017	[64]
	**	0.807 (+7.03%)	20.233 (−19.13%)	
Corn gluten meal	*	0.178	8.387	[68]
	**	0.191 (+7.30%)	6.194 (−26.15%)	
Wheat gluten	*	0.415	45.233	[63]
	**	0.427 (+2.89%)	38.243 (−15.45%)	
Sunflower meal protein	*	0.500	7.880	[44]
	**	0.510 (+2.00%)	6.990 (−11.29%)	
Cellulase	*	1.680	49.650	[90]
	**	1.75 (+4.17%)	43.480 (−12.43%)	
Mulberry leaf	*	0.767	41.336	[89]
	**	0.805 (+5.02%)	35.522 (−14.07%)	
Silkworm pupa protein	*	0.209	11.835	[86]
	**	0.244 (+16.75%)	11.078 (−6.396%)	

* Control (without ultrasound); ** pretreated by ultrasound. Values in parentheses indicate the increase or decrease (with negative sign) in enzymolysis kinetics (K_A_ and K_M_).

**Table 3 foods-12-04027-t003:** Effect of ultrasonic pretreatment on the enzymatic thermodynamic parameters.

Substrate	Type	E_a_ (kJ/mol)	ΔH (kJ/mol)	ΔS (J/mol·K^−1^)	ΔG (kJ/mol)	Reference
Corn gluten meal	*	49.07 ± 1.12	46.63 ± 1.12	−133.83 ± 3.82	85.86	[68]
	**	37.78 ± 0.93 (−23.00%)	35.34 ± 0.93 (−24.21%)	−167.23 ± 3.17 (−24.96%)	84.37 (−1.74%)	
Egg white protein	*	86.70 ± 1.10	84.10 ± 1.80	−226.80 ± 3.80	156.20	[78]
	**	33.20 ± 0.80 (−61.71%)	30.60 ± 0.80 (−63.61%)	−236.60 ± 2.80 (−4.32%)	105.80 (−32.27%)	
Whey protein	*	46.92 ± 2.12	44.37 ± 1.87	−140.74 ± 4.76	87.74	[79]
	**	39.46 ± 1.57 (−15.89%)	36.90 ± 0.99 (−16.83%)	−163.87 ± 3.29 (−16.43%)	87.40 (−0.39%)	
Sodium caseinate	*	46.39 ± 0.03	43.65 ± 0.04	−141.55 ± 0.21	90.10	[83]
	**	33.42 ± 0.02 (−27.96%)	30.93 ± 0.13 (−29.14%)	−179.71 ± 0.12 (−26.96%)	89.55 (−0.61%)	
Tea residue protein	*	46.93 ± 0.47	44.24 ± 0.03	99.69 ± 0.70	30.72	[82]
	**	42.95 ± 0.62 (−8.48%)	40.26 ± 0.58 (−9.00%)	107.19 ± 1.55 (+7.52%)	33.23 (+8.17%)	
Rapeseed protein	*	39.66 ± 1.71	37.16 ± 1.23	−192.87 ± 3.04	93.68	[87]
	**	27.94 ± 1.59 (−29.56%)	25.35 ± 1.37 (−31.79%)	−228.80 ± 2.58 (−18.63%)	92.39 (−1.37%)	
Zein	*	48.55 ± 1.97	46.05 ± 0.99	−154.47 ± 3.02	93.63	[43]
	**	39.06 ± 1.21 (−19.52%)	36.55 ± 1.35 (−20.63%)	−163.98 ± 2.98 (−6.16%)	87.06 (−7. 02%)	
Sunflower meal protein	*	31.51 ± 0.38	28.93 ± 0.38	−215.95 ± 1.24	92.20	[44]
	**	23.86 ± 0.29 (−24.28%)	21.37 ± 0.28 (−26.13%)	−237.56 ± 0.92 (−9.10%)	91.00 (−1.30%)	
Dextransucrase	*	37.74 ± 2.41	35.29 ± 2.42	−140.74 ± 8.18	77.93	[91]
	**	25.15 ± 0.89 (−33.7%)	22.69 ± 0.58 (−35.7%)	−181.33 ± 1.96 (−28.8%)	77.63 (−0.38%)	
Mulberry leaf	*	29.81 ± 1.84	27.29 ± 1.84	−193.11 ± 6.12	83.30	[89]
	**	16.55 ± 1.68 (−44.48%)	14.08 ± 1.68 (−48.41%)	–233.89 ± 5.45 (−21.12%)	82.60 (−0.84%)	

* Control (without ultrasound); ** pretreated with ultrasound; E_a_—activation energy; ΔH—change in enthalpy; ΔS—entropy of activation; ΔG—Gibbs free energy. Values in parentheses indicate an increase or decrease (with negative sign) in the thermodynamic kinetics (E_a_, ΔH, ΔS, and ΔG).

**Table 4 foods-12-04027-t004:** Effect of ultrasonic pretreatment conditions on enzymolysis.

Substrate	Ultrasonic Condition	Results	Reference
Cellulose	20 kHz; 535 W; 25 °C; 2 and 4 h	The longer the exposure time of cellulose to ultrasound, the greater the swelling and decrease in Segal CI. The molecular size and surface area of cellulose increase with time.	[92]
Citrus pectin	22 kHz; 900 W; 20 °C; 0–27 W mL^−1^	With an increase in the ultrasound intensity from 0 to 18 W mL^−1^, the molecular weight of pectin decreased significantly (from 485.10 kDa to 240.11 kDa).	[84]
Whey protein	20 kHz; 0–500 W; 25 °C; 0–20 min	With ultrasonic power and time increasing, the DH value initially increased to a point and then decreased gradually.	[79]
Sugar beet shreds	22–25 kHz; 540 W; 20 and 30 min	Compared with the other conditions, under 540 W, 20 min, a 66.7% duty cycle, and two solid conditions, the reducing sugar yield obtained by enzymolysis was the highest, about 780 mg/g of cellulose.	[55]
Wheat gluten powder	150 W L^−1^; 30 °C; 15 min	Alternating dual-frequency-ultrasound-assisted enzymolysis is better than simultaneous dual-frequency ultrasound, which may be due to the reduction in cavitation bubbles caused by the superposition of sound waves under dual-frequency ultrasound mode.	[66]
Rice protein	27.3 kHz; 120 W L^−1^; 50 °C	Random sweep frequency ultrasound pretreatment can significantly increase the inhibitory activities of DH and ACE and improve the enzymatic hydrolysis effect.	[70]
Rapeseed protein	20 kHz; 0–1200 W; 50 °C; 0–18 min	Compared with other UP conditions, a maximum DH of 22.07% and ACE inhibitory activity of 72.13% were achieved at 600 W with 12 min of pretreatment.	[93]
Wheat gluten	20/35 kHz; 0–300 W L^−1^; 0–25 min	The surface roughness of wheat protein was different under alternating dual-frequency UP with different durations and power intensities, and the change in the degree of the secondary conformation was different.	[77]

**Table 5 foods-12-04027-t005:** Effects of ultrasound treatment on the secondary structures of enzymes.

Enzyme	Condition	α-Helix (%)	β-Sheet (%)	β-Turn (%)	Random Coil (%)	Activity	Reference
Pectinase	Control	2.60	40.94	19.42	37.04	31.94%	[104]
	Sonicated	2.70	41.06	19.38	36.86		
Pectinase	Control	2.60	40.86	19.48	37.06	20.41%	[105]
	Sonicated	2.70	41.14	19.22	36.94		
Alcalase	Control	77.00	1.00	-	22.00	5.81%	[99]
	Sonicated	81.00	0	-	19.00		
Dextranase	Control	19.70	24.10	19.20	37.00	13.57%	[106]
	Sonicated	22.80	23.50	18.70	35.00		
Cellulase	Control	26.20	26.60	21.90	24.80	18.17%	[40]
	Sonicated	23.40	25.00	23.70	32.10		
Glucoamylase	Control	13.50	13.30	22.60	42.70	27.52%	[107]
	Sonicated	15.90	10.60	21.70	48.00		
Endoglucanase	Control	31.10	28.70	-	-	5.37%	[108]
	Sonicated	31.20	28.60	-	-		
Cellulase	Control	7.05	36.10		56.85	19.05%	[90]
	Sonicated	1.85	39.00		59.15		

**Table 6 foods-12-04027-t006:** Application of ultrasound enzyme extraction/hydrolysis in food processing.

Application	Case	Reference
Producing bioactive peptides	Produce zein peptides with high ACE inhibitory activity	[43]
	Produce wheat gluten peptides with high ACE inhibitory activity	[66]
	Produce antioxidant peptides from corn	[129]
	Produce antioxidant peptides from potato protein	[124]
	Extract polypeptides from Cordyceps militaris	[125]
Improving the quality of the product	Increase the contents of slow digestible starch and resistant starch in pea starch	[131]
	Improve the solubility, foaming, and emulsifying properties of egg white protein	[132]
	Enhance the gelation property of wheat gluten	[133]
	Enhance the hydrophobic and antioxidant properties of soy protein isolate hydrolysate	[134]
Brewing red wine and developing juice drinks	Optimize the process of red wine impregnation and extract phenols and volatile compounds	[135]
	Enhance the yield, clarity, and total soluble solids content of banana juice	[136]
	Enhance the extraction of compounds and chromatic properties of mulberry	[123]
Assisting with determining the hazardous heavy metals content in foods	Assist with determining the content of Cd in rice flour	[137]
	Assist with determining the content of Mn in onion, parsley, chili powder, black pepper, and tomato samples	[138]
	Assist in determining the content of arsenic in rice and flour samples	[139]
Reusing food waste	Extract polyphenols from pomegranate peel waste	[127]
	Extract lycopene pigment from tomato processing waste	[140]
	Extract high-quality pectin from sisal waste	[42]
	Produce ellagic acid from blueberry pulp	[141]
	Produce pectin from citrus processing waste	[126]
	Isolate β-glucan from oat bran	[142]
Synthetizing food additives	Hydrolysis of aspirin to methyl salicylate	[143]
	Catalyze transesterification for the synthesis of hexyl acetate	[144]
	Catalyze transesterification for the synthesis of cinnamyl acetate	[145]

## Data Availability

The data used to support the findings of this study can be made available by the corresponding author upon request.

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
