# Peer review of "Ultrasound-Assisted Enzymatic Protein Hydrolysis in Food Processing: Mechanism and Parameters"

_foods, 2023, doi:10.3390/foods12214027_

Round 1
Reviewer 1 Report
Comments and Suggestions for Authors
Overall, the manuscript provides a comprehensive overview of ultrasound-assisted enzymatic hydrolysis (UES) and its mechanisms. It discusses the effects of ultrasound on both the structure of enzymes and substrates, as well as the key parameters influencing UES. Additionally, it highlights various applications of UES in food processing.
Here are some comments and suggestions to improve the manuscript:
- Clarity and Organization:
- The manuscript could benefit from better organization and clearer section headings to make it easier for readers to follow the content.
- Consider providing an introductory section that outlines the purpose and scope of the manuscript to give readers a clear roadmap of what to expect.
- Citations and References:
- Ensure that all references are properly cited within the text. Some citations appear to be missing corresponding in-text citations.
- Check the formatting of the citations and references to ensure consistency (e.g., use a consistent citation style, such as APA or IEEE).
- Figures:
- Figure captions should be more descriptive. They should provide a clear understanding of what each figure represents without the need to refer to the text.
- Figure 4 is mentioned in the text, but the figure itself is missing in the manuscript. Make sure all figures are included.
- Language and Grammar:
- There are some grammatical issues and awkward sentences in the manuscript that need revision for clarity. For example, in "The disadvantage of traditional extraction technology to the ecological environment led to the emergence of this new extraction technology," consider rephrasing for clarity.
- Subheadings:
- Use consistent formatting for subheadings throughout the manuscript. Ensure that subheadings are easy to distinguish from the main headings.
- Tables:
- Table 6 appears to have formatting issues, making it difficult to read. Ensure proper alignment of data in tables.
- Future Directions:
- It would be beneficial to include a section or paragraph at the end of the manuscript discussing potential future directions or areas for further research in the field of UES.
- Conclusion:
- The conclusion section could be expanded to provide a more comprehensive summary of the key findings and their implications.
- Length and Detail:
- Consider whether some sections could benefit from more detailed explanations or additional examples to enhance understanding.
- Citation of Recent Research:
- Ensure that the manuscript cites recent research in the field, as the field of ultrasound-assisted enzymatic hydrolysis is continually evolving.
- Review of Typos and Errors:
- Carefully review the manuscript for typographical errors, missing words, and other minor issues that can impact readability.
- Equations and Symbols:
- If there are equations or symbols used in the text, ensure that they are properly defined or explained for clarity.
Comments on the Quality of English Language
no issues
Reviewer 2 Report
Comments and Suggestions for Authors
The paper deals with the use of ultrasound for protein hydrolysis.
In general it is well structured, but the wording needs to be revised, as the reading is sometimes a bit clumsy and not very fluent.
In l10 it is not clear why the word physiotherapy is used.
In l18 it talks about the state of ultrasonic device, it should be devices?
In the introduction you could go into a little more detail about the main applications of ultrasound in food.
It would be interesting if the article developed in more detail aspects of ultrasound generation (cavitation, implosion, explosion).
